# Effect of Resistance Exercise Order on Cardiovascular Disease Risk Factors in Older Women: A Randomized Controlled Trial

**DOI:** 10.3390/ijerph20021165

**Published:** 2023-01-09

**Authors:** Crisieli M. Tomeleri, Paolo M. Cunha, Márcia M. Dib, Durcelina Schiavoni, Witalo Kassiano, Bruna Costa, Denilson C. Teixeira, Rafael Deminice, Ricardo José Rodrigues, Danielle Venturini, Décio S. Barbosa, Cláudia R. Cavaglieri, Luís B. Sardinha, Edilson S. Cyrino

**Affiliations:** 1Metabolism, Nutrition, and Exercise Laboratory, State University of Londrina, Londrina 86057-970, PR, Brazil; 2Department of Pathology, Clinical and Toxicological Analysis, State University of Londrina, Londrina 86057-970, PR, Brazil; 3Faculty of Physical Education, University of Campinas, Campinas 13083-970, SP, Brazil; 4Exercise and Health Laboratory, CIPER, Faculdade de Motricidade Humana, Universidade de Lisboa, 1649-004 Lisboa, Portugal

**Keywords:** strength training, intensity, volume, oxidative stress, inflammatory biomarkers, women’s health

## Abstract

We compared the effects of two specific resistance training (RT) exercise orders on cardiovascular risk factors. Forty-four untrained older women (>60 years) were randomly assigned to three groups: control (CO*N*, *n* = 15), multi-joint to single-joint (MJ-SJ, *n* = 14), and single-joint to multi-joint (SJ-MJ, *n* = 15) exercise orders. Training groups performed a whole-body RT program (eight exercises, 3 × 10–15 repetitions for each exercise) over 12 weeks in 3 days/week. Body fat, triglycerides, total cholesterol, HDL-c, LDL-c, VLDL-c, glucose, IL-6, IL-10, TNF-α, C-reactive protein, total radical-trapping antioxidant (TRAP), advanced oxidation protein products (AOPP), ferrous oxidation-xylenol (FOX), and nitric oxide concentrations (NOx) were determined pre- and post-intervention. Significant interaction group × time (*p* < 0.05) revealed reducing fat mass and trunk fat and improvements in glucose, LDL-c, IL-10, TNF-α, C-reactive protein, FOX, and AOPP concentrations in both training groups, without differences between them (*p* > 0.05). The results suggest that 12 weeks of RT, regardless of exercise order, elicit positive adaptations on body fat and metabolic biomarkers similarly in older women.

## 1. Introduction

Cardiovascular diseases are the main cause of worldwide mortality, affecting a large part of the older population, with high prevalence in women [1] since the cardiovascular disease risk factors (e.g., C-reactive protein, triglycerides, LDL-c) appear to be exacerbated by the menopause [2,3,4]. In contrast, a positive effect of resistance exercises on cardiovascular disease risk factors in older adults was reported recently in some studies [5,6]. In this regard, resistance training (RT) has been recommended as an essential non-pharmacological strategy to attenuate these age-related impairments in health status [7]. The main benefits of RT practice in older women include an increase in muscle mass and muscular strength, a reduction of body fat, improvement in the lipid profile, oxidative stress, and inflammatory biomarkers [8].

However, RT-related changes magnitude may be affected by the manipulation of the variables that compose training programs, which are related to intensity (e.g., perceived effort, external load, the velocity of muscle action, and rest intervals), volume (e.g., number of exercises, sets, repetitions, and frequency) and structure (e.g., exercise selection and exercise order). Considering that the exercise order may affect the intensity and volume of a training session, this variable may play a crucial role in the RT-induced adaptative responses [9,10,11]. Current guidelines suggest performing a training session with an exercise order that initiates with the multiple-joint (MJ) exercises—i.e., involving the dynamic participation of two or more joints—before single-joint (SJ)—i.e., involving the dynamic participation of one specific joint [7]. Some researchers believe that this sequence may induce better adaptations than opposite exercise orders because it allows the application of higher training loads [7,11,12].

Nevertheless, current recommendations were mainly based on data from acute investigations [7]. In addition, the few chronic studies on exercise order usually focus on body composition, muscular strength, and functional fitness [11,13,14,15]. Therefore, less is known regarding the potential influence of exercise order on other outcomes, such as cardiovascular disease risk factors. Only one study investigated the impact of RT performed in exercise orders different on health parameters in the older population [16]. Although the study has found exciting results, regardless of exercise order, the sample was composed of trained women and did not include a control group.

Recent studies revealed that the exercise order might also affect the volume load (repetitions × load) [9,11], a variable that seems to be an important mediating factor of the adaptations elicited by RT [17,18]. Furthermore, given that muscle contraction acts as an endocrine organ [19], if a given exercise order allows training with higher loads in exercises that involve a greater amount of muscle mass (i.e., MJ), theoretically, it is plausible to believe that the acute and chronic effects on inflammatory markers could be optimized. As a consequence of reducing inflammatory biomarkers, such as C-reactive protein, the chance of lowering possible cardiovascular complications increases [2]. However, studies are needed to test such a hypothesis. 

Therefore, this study aimed to compare the effects of two specific exercise orders (MJ-to-SJ versus SJ-to-MJ) on cardiovascular disease risk factors. We hypothesized that the training protocol with the MJ-SJ exercise order would induce greater improvements in body fat, glucose, lipid profile, inflammatory, and oxidative stress biomarkers because of the potential higher volume load training achieved with this exercise order.

## 2. Materials and Methods

### 2.1. Experimental Design

This work is part of the “Active Aging Longitudinal Study”, a research project initiated in 2012 to investigate the effects of supervised, structured, and progressive RT programs on neuromuscular, morphological, physiological, metabolic, cognitive, and behavioral outcomes in older women. The sample in the present study consisted of the 2015 annual cohort of participants from this research project. A randomized controlled trial was carried out over 16 weeks, with 12 weeks dedicated to the RT program and four weeks to assessments. Anthropometric, muscular strength, body fat measurements, and venous blood collections for biochemical analysis were performed in weeks 1–2 and 15–16 pre- and post-intervention. The training groups performed a supervised progressive RT program between weeks 3–14. The control group did not perform any exercise during this period. This investigation is a secondary analysis of the previous work published [11].

### 2.2. Participants

Participant recruitment was carried out through newspaper and radio advertising. A preliminary sample was selected through an interview and clinical anamnesis. All participants completed health history and physical activity questionnaires and met the following inclusion criteria: female, >60 years, physically independent, free from cardiac or orthopedic dysfunction, not receiving hormonal replacement therapy, and not performing any regular physical exercise more than once a week over the six months preceding the onset of the current experiment. A diagnostic graded exercise test with a 12-lead electrocardiogram was conducted on the sample selected for this study. The exams were reviewed by a cardiologist, and they were released without restrictions to participate in this study. Participants of training groups who did not reach adherence ≥ 85% of training sessions were excluded from the analysis.

A priori sample analysis revealed that to achieve an effect size = 0.5 for muscular strength with a power of 0.8 and an alpha error = 5%, a total of 42 participants would be necessary (14 for each group). Considering a drop-out rate of approximately 25%, we recruited 54 older women who were randomly assigned to one of three conditions: a control group that did not perform any exercise (CO*N*, *n* = 18) and two training groups that performed the RT program in a multi-joint to single-joint order (MJ-SJ, *n* = 18), or in a single-joint to multi-joint order (SJ-MJ, *n* = 18). Written informed consent was obtained from all participants after receiving a detailed description of the study procedures. This investigation was conducted according to the Declaration of Helsinki and was approved by the local University Ethics Committee.

### 2.3. Fat Mass and Trunk Fat

Dual-energy X-ray absorptiometry exams (DXA) were conducted in a Lunar Prodigy device, model NRL 41990 (GE Lunar, Madison, WI, USA), to determine the fat mass and trunk fat. Information on lean soft tissue was previously reported in another study [11]. Participants were submitted to the exams wearing light clothes, barefoot, and without any metallic object or other accessory items on their bodies. A laboratory technician with experience in this type of evaluation performed both calibration and analysis. Previous test-retest scans of 12 older women measured 24–48 h apart resulted in a standard error of measurement (SEM) of 0.10 kg for body fat and 0.67 kg for trunk fat, with an intraclass correlation coefficient (ICC) >0.99 for all variables.

### 2.4. Metabolic Biomarkers

Venous blood samples were collected into one tube between 7:00 am and 9:00 am after a 12 h fast and at least 48 h since the last RT session. Five milliliters were withdrawn from a prominent superficial vein in the antecubital space using a clean venous puncture with minimal stasis and placed in a tube containing a dipotassiumethylenediaminetetra-acetic acid (EDTA) as an anticoagulant and preservative. All samples were centrifuged at 3000 rpm for 15 min, and plasma or serum aliquots were stored at −80 °C until assayed. Inter- and intra-assays coefficients of variation were <10% as determined in human plasma. Glucose, total cholesterol (TC), high-density lipoprotein (HDL-c), triglycerides (TG), and high-sensitivity C-reactive protein concentrations were determined by standard methods in a specialized laboratory at University Hospital. The LDL-c was calculated using the following equation: LDL-c = TC − (HDL-c) − (TG/5), with TG/5 = VLDL-c. The analyses were performed using a biochemical auto-analyzer system Dimension RxL Max (Siemens Healthcare Diagnostics, Malvern, PA, USA) according to established methods in the literature consistent with the manufacturer’s protocol. Interleukin-10 (IL-10), Interleukin-6 (IL-6), and TNF-α were determined by enzyme-linked immunosorbent assay (ELISA), according to the specifications of the manufacturer (R&D Systems, Minneapolis, MN, USA). Advanced oxidation protein products (AOPP) were determined in the plasma using a semiautomatic method described by Witko-Sarsat et al. [20]. The total radical-trapping antioxidant parameter (TRAP) was determined according to Repetto et al. [21]. This method detects hydrosoluble and/or liposoluble plasma antioxidants by measuring the chemiluminescence inhibition time induced by 2,2-azobis (2-amidinopropane). The system was calibrated with the vitamin E analog Trolox^®^. Plasma lipid-hydroperoxides levels were determined by ferrous oxidation-xylenol orange (FOX) assay [22]. Serum nitric oxide metabolites (NOx) concentrations were assessed by nitrite (NO2-) and nitrate (NO3-) concentration according to the Griess reaction, supplemented by the reduction of nitrate to nitrite with cadmium [22].

### 2.5. Dietary Intake

Food intake was assessed by the 24 h dietary recall method applied on two non-consecutive days of the week, with a photographic record taken during an interview. Dietary intake was monitored in the first two and the last two weeks of the intervention period. The homemade measurements of the nutritional values of food were converted into grams and milliliters by the online software Virtual Nutri Plus (Keeple^®^, Rio de Janeiro, RJ, Brazil) for diet analysis. Some foods were not found in the program database. Therefore, these items were added from food tables.

### 2.6. Muscular Strength

Maximal dynamic strength was evaluated through the one-repetition maximum (1RM) test on the chest press, leg extension, and preacher curl exercises. The participants completed three attempts in each exercise for three testing sessions [23]. The three 1RM testing sessions were conducted in the morning, each one separated by 48 h of recovery. The rest period was 3–5 min between each attempt and five minutes between exercises. The 1RM was recorded as the higher resistance lifted in which the participant could complete only one single maximal repetition. Muscular strength was determined by the sum of each 1RM in the three exercises. The ICC from our lab for these tests is ≥0.96 with an SEM ≤2.0 kg.

### 2.7. Resistance Training Program

Participants from both training groups (MJ-SJ and SJ-MJ) performed the same 12-week RT program with eight exercises. All participants were personally supervised by physical education professionals with substantial RT experience to help ensure consistent and safe exercise performance, carried out differing only in the exercise order. The RT sessions were performed three times per week on Mondays, Wednesdays, and Fridays during the morning hours. Participants of the MJ-SJ group performed exercises in the following order: chest press, seated row, triceps pushdown, preacher curl, horizontal leg press, leg extension, leg curl, and seated calf raise. On the other hand, participants of the SJ-MJ group performed exercises in the following order: preacher curl, triceps pushdown, seated row, chest press, seated calf raise, leg curl, leg extension, and horizontal leg press. The RT sessions consisted of 3 sets of 10–15 repetitions per exercise; if the participant reached momentary muscle failure—i.e., attempted to perform another repetition with proper technique but was unsuccessful—before the upper limit of repetitions, the load was kept the same for the next session. On the other hand, when 15 repetitions were completed in the three sets for a determined exercise, the load was adjusted by 2–10% [7]. Participants rested for 1 to 2 min between each set and for 2 to 3 min between each exercise. Instructors adjusted loads of each exercise according to the participant’s ability and improved exercise capacity throughout the study to ensure that they were exercising with as much resistance as possible while maintaining proper exercise technique. Training load (kg) and the number of sets and repetitions performed in all exercises were individually recorded. The volume load of each exercise during each session was calculated by the following equation: exercise volume load = load (kg) × sets (no.) × repetitions (no.). The volume load of each week was calculated by summing volume load of the eight exercises obtained in each training session.

### 2.8. Statistical Analyses

One-way analysis of variance (ANOVA) was used to compare general characteristics and weekly VL. A Chi-square test was used to compare clinical characteristics. Generalized Estimating Equations (GEE) were applied for pre- to post-intervention and inter-group comparisons. When the Wald χ2 was significant, a Fisher post hoc test was employed to identify the mean differences. The ES was calculated as the post-training mean minus pre-training mean divided by the pooled standard deviation [24]. An ES of 0.00–0.19 was considered trivial, 0.20–0.49 was considered small, 0.50–0.79 was moderate, and ≥0.80 was large. Significance was accepted at *p* <0.05. The data were stored and analyzed using IBM SPSS Statistics, v. 24.0 (IBM Corp., Armonk, NY, USA).

## 3. Results

Forty-four participants finalized the experimental protocol (MJ-SJ = 15, SJ-MJ = 14, and CO*N* = 15). Withdrawals occurred due to personal reasons (MJ-SJ = 2; SJ-MJ = 2, CO*N* = 2), and surgery (MJ-SJ = 1; SJ-MJ = 2, CO*N* = 1). Training attendance did not differ between groups (MJ-SJ: 98.8% and SJ-MJ: 96.3%; *p* > 0.05). The participants’ general characteristics and medical history are presented in supplemental digital content 1. There was no significant difference between groups for general characteristics at baseline, medical history, and treatment (*p* > 0.05).

### 3.1. Volume Load

The volume load for each week and according to the group are presented in Figure 1. The MJ-SJ and SJ-MJ groups exhibited greater values of the volume load in weeks 4–7 and 9–11 compared to week 1, however, without differences between training groups (*p* < 0.05).

### 3.2. Dietary Intake and Main Outcomes

Table 1 presents pre- and post-training results for dietary intake, muscular strength, body fat, glucose, lipid profile, inflammatory, and oxidative stress markers. Daily energy and macronutrient intake were not different between groups and did not change over time (*p* > 0.05). Significant time × group interaction (*p* < 0.05) revealed RT effects in both training groups for fat mass, trunk fat, glucose, LDL-c, IL-10, TNF-α, C-reactive protein, FOX, and AOPP when compared to the control group. No significant differences were observed before to after intervention in any group for TC, HDL-c, VLDL-c, TG, IL-6, TRAP, and NOx (*p* > 0.05). A significant interaction time × group (*p* < 0.05) was found for muscular strength with higher values after the intervention for both training groups when compared to the control group (MJ-SJ, ES = +1.0; SJ-MJ, ES = +0.62; CO*N*, ES = −0.51).

Table 2 presents the effect-size values for the variables analyzed. Changes in body fat and the lipid profile biomarkers were of trivial–small magnitude for training groups. In contrast, the glucose changes were large, whereas inflammatory and oxidative stress biomarkers were of moderate and small magnitude, respectively.

## 4. Discussion

The main findings of this study were that RT was effective for decreasing fat mass and trunk fat, as well as for improving glucose concentrations and inflammatory (IL-10, TNF-α, C-reactive protein) and oxidative stress biomarkers (AOPP and FOX), regardless of the exercise order. Our hypothesis that the MJ-SJ group would present superior responses was refuted. According to our knowledge, this is the first study comparing the effects of different exercise orders on cardiovascular disease risk factors in untrained older women. Because of this unique approach, comparisons with other studies are limited. It is worth highlighting that we found similar gains in muscular strength and muscle mass for MJ-SJ and SJ-MJ exercise orders in this same sample [11].

Given that the MJ-SJ order could favor the accumulation of a higher volume load, we hypothesized that this order could be more efficient in improving risk factors for cardiovascular diseases. The volume load is a determining variable for the RT adaptations [7]. However, we have found a similar volume load between MJ-SJ and SJ-MJ (see Figure 1), and we found a similar reduction in cardiovascular risk to MJ-SJ and SJ-MJ over 12 RT weeks. In this context, such similarities can be explained, at least in part, through the intensity of effort. In this regard, Ribeiro et al. [25] observed that older women experience a similar perception of effort when training MJ to SJ and SJ to MJ. There are indications that the RT adaptations can be determined by the intensity/perception of effort [26,27,28]. Therefore, it is possible that the similarities observed in the present study may be related to the similar intensity of effort, which was probably matched between the groups. Importantly, although plausible, we do not measure the perceived effort, so this assertion needs further investigation.

For IL-6, although the effect of RT was not significant for both training groups, effect size values indicated that the decrease was of moderate magnitude. In contrast, the control group had an increase in the IL-6 concentration. As it is a pro- and anti-inflammatory cytokine, the IL-6 responses, by itself, do not allow us to state that the increase or decrease was due to corresponding to a less inflamed environment [29]. In this regard, evaluating other pro- and anti-inflammatory cytokines is necessary to improve the interpretation of results. In our study, both training groups showed a moderate reduction in IL-6 and a decrease in TNF-α, CRP, and an IL-10 increase. In contrast, although it has not shown a significant increase in IL-6, the control group showed an increase in pro-inflammatory cytokines and a reduction in anti-inflammatory ones. These results indicate that RT improves the inflammatory profile, disregarding the exercise order used. These findings are important because they suggest RT’s reversal or deceleration effects of RT on the deleterious alterations associated with the aging process, i.e., the presence of a chronic low-grade inflammation increases the risk of developing age-associated diseases and is an independent risk factor for mortality [30,31].

We also observed a reduction in blood glucose and LDL-c in MJ-SJ and SJ-MJ contrasted to the control group. The lack of investigations that directly compare the effect of the exercise order on blood markers restricts us from comparing with previous findings; notwithstanding, our results are in accordance with previous studies that found favorable effects of RT on these outcomes in older women [6,32,33,34]. Some factors may explain these changes in blood glucose and LDL-c. Previous studies observed that reducing body fat could modify lipid profile and blood glucose [6,34]. The most significant reductions appear in the android region [35]. Thus, the changes in body fat found in our study, especially trunk fat, can help explain, at least in part, the modification in LDL-c and glucose. The relationship between body fat and lipid profiles may be associated with insulin resistance and increased free fatty acids, a condition leading to the formation of large TG-rich VLDL particles, which alters the expression of key enzymes in the plasma, such as decreased lipoprotein lipase [36]. Additionally, the changes mentioned above can be due to an increase in the ability of the skeletal muscle to use fat and glucose, and an improvement in insulin sensitivity, thereby reducing the levels of plasma lipids and blood glucose [8]. Although plausible, these findings are not universal, and some studies have not reported RT’s effect on lipoproteins [37] or blood glucose [38]. Discrepancies across results are not clear but may be related to factors such as initial lipid levels and glucose, gender, age, body fat, duration, and intensity of the training program. Besides, the responsiveness of the lipid profile induced by RT depends on genotype [8], raising the likelihood that genetic factors ultimately determine the magnitude to which RT affects the lipid profile. 

Regarding oxidative stress, we did not observe any difference between the two exercise orders; both intervention groups improved oxidative stress biomarkers (i.e., AOPP and FOX). Few studies have investigated the RT effect—and consequently the exercise order—on oxidative stress biomarkers. From the available findings, it is possible to notice some divergences in the impact of RT on these biomarkers. For example, Ribeiro et al. [39] revealed that eight weeks of RT performed three times a week with three sets per exercise of 8–12 RM were sufficient to improve oxidative stress by maintaining the body’s antioxidant capacity (TRAP) while decreasing protein oxidation (AOPP). On the other hand, Padilha et al. [40] showed that 12 weeks of RT performed 2 or 3 days/week with 1 set per exercise of 10–15 RM improved oxidative stress by increasing TRAP while maintaining AOPP. Furthermore, our study revealed significant changes in FOX in both groups, whereas Ribeiro et al. [39] did not find any difference in FOX following RT. Although there is no definitive mechanism to support such divergences, it is feasible to speculate that differences between training protocols, such as training volumes, intensity, and duration of intervention, may have influenced adaptations. The physiological mechanisms by which RT improves oxidative stress may be attributed to some factors. Physical exercise can increase the synthesis of reactive oxygen and nitrogen species by activating the electron transport chain and the synthesis of lactic acid, catecholamines, and inflammatory factors that contribute to the production of reactive oxygen species in mitochondria [41]. Moreover, anaerobic exercise, such as RT, may increase the synthesis of xanthine oxidase and NADPH oxidase enzymes, and both influence oxidative stress modulation [41]. Thus, the antioxidant system adapts in response to this process, with adjustments favorable to the endogenous antioxidant system, improving the body’s defense capacity. 

It is essential to feature the strengths of the present study. Professionals with RT experience supervised all training sessions to ensure participant safety and the proper exercise technique. The importance of supervised RT in older adults has been reported in some studies [42,43]. Moreover, the weight adjustments were continuous and based on the participant’s progress throughout the RT sessions, allowing the optimal stimulus to be maintained throughout the intervention. Also, monitoring the participants’ dietary intake allowed a more consistent analysis of RT effects. Despite the recognized limitations associated with using the 24 h dietary recall method to intake assessment, particularly a tendency for individuals to underestimate dietary intake, no difference was observed within- and between-group. Finally, the blood biomarkers analyzed in this study are essential for screening the health of older adults and can be modified with RT, independent of the exercise order. Conversely, it is worthwhile to note that this investigation has some limitations. First, 12 weeks of intervention can be considered a comparatively short time. Thus, it is necessary to determine whether results would differ over a longer training period. Although magnetic resonance imaging and computed tomography are the methods recommended for visceral fat evaluation, with the introduction of visceral fat assessment software for DXA scanners, DXA’s use for this purpose has become quite common due to its low cost, brief examination time, noninvasiveness, limited radiation exposure, and satisfactory level of precision [44]. However, caution should be observed when contrasting our results with other studies that used more consistent body fat measurements. Finally, the study is specific to untrained older women. Therefore, results should not be generalized to other populations, including participants of different age groups and initial fitness levels, as different adaptative responses to RT may be influenced by such factors.

## 5. Conclusions

Our results suggest that 12 weeks of RT in both MJ-SJ and SJ-MJ exercise orders seem effective for improving body fat and blood biomarkers in untrained older women. Therefore, professionals and researchers can choose exercise order of preference between MJ-SJ and SJ-MJ when prescribing RT exercises focused on inducing adaptations in body fat and blood biomarkers in physically independent older women.

## Figures and Tables

**Figure 1 ijerph-20-01165-f001:**
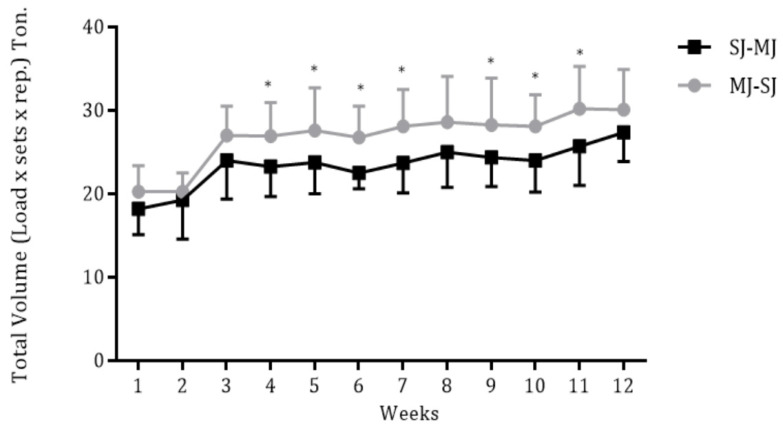
Weekly volume load according to training groups (MJ-SJ = 15 and SJ-MJ = 14). * *p* < 0.05, significant difference when compared to week 1. MJ-SJ = group that performed RT in a multi-joint to single-joint order, SJ-MJ = group that performed RT in a single-joint to multi-joint order.

**Table 1 ijerph-20-01165-t001:** Dietary intake, total strength, body fat, and blood biomarkers after the 12-week period of intervention.

Measures	MJ-SJ (*n* = 15)		SJ-MJ (*n* = 14)		CO*N* (*n* = 15)	
	Pre	Post	Pre	Post	Pre	Post
**Dietary intake**						
Energy (kcal/kg)	16.1 ± 3.4	16.2 ± 3.9	16.2 ± 3.9	16.7 ± 3.1	16.1 ± 3.2	16.4 ± 3.5
Carbohydrates (g/kg)	2.3 ± 0.7	2.3 ± 0.6	2.6 ± 0.9	2.5 ± 0.7	2.5 ± 0.7	2.3 ± 0.9
Proteins (g/kg)	0.8 ± 0.3	0.8 ± 0.3	0.8 ± 0.3	0.9 ± 0.2	0.8 ± 0.3	0.9 ± 0.4
Lipids (g/kg)	0.5 ± 0.2	0.5 ± 0.2	0.5 ± 0.2	0.5 ± 0.2	0.5 ± 0.3	0.6 ± 0.3
**Total strength (kg)**	114.1 ± 17.0	132.9 ± 21.5 ^§^	118.4 ± 22.1	130.9 ± 19.0 ^§^	111.6 ± 14.2	104.4 ± 14.7
**Body fat**						
Fat mass (kg)	27.3 ± 8.6	25.9 ± 9.1 *^§^	25.4 ± 7.2	24.0 ± 7.3 *^§^	26.4 ± 8.7	26.9 ± 8.6
Relative body fat (%)	41.8 ± 6.8	39.0 ± 7.4 *	40.0 ± 5.5	37.7 ± 5.5 *^§^	41.1 ± 7.4	41.7 ± 7.1
Trunk fat (kg)	13.7 ± 4.2	13.1 ± 4.4 *^§^	12.5 ± 3.6	11.6 ± 3.4 *^§^	14.9 ± 4.8	15.0 ± 4.6
Relative trunk fat (%)	42.6 ± 6.7	39.9 ± 7.6 *^§^	39.9 ± 6.5	37.1 ± 5.9 *^§^	42.9 ± 6.1	43.2 ± 5.4
**Blood biomarkers**						
Glucose (mg/dL)	112 ± 19	91 ± 16 *^§^	114 ± 17	88 ± 12 *^§^	100 ± 11	102 ± 10
TC (mg/dL)	189 ± 26	190 ± 16	193 ± 38	183 ± 32	203 ± 20	214 ± 21
HDL-c (mg/dL)	53 ± 10	54 ± 12	54 ± 14	56 ± 15	55 ± 12	53 ± 11
LDL-c (mg/dL)	115 ± 23	112 ± 18 ^§^	118 ± 40	105 ± 32 *^§^	124 ± 24	139 ± 21 *
VLDL-c (mg/dL)	24 ± 10	21 ± 9	22 ± 10	22 ± 8	22 ± 7	23 ± 9
Triglycerides (mg/dL)	107 ± 28	119 ± 38	108 ± 40	109 ± 53	117 ± 46	110 ± 37
IL-6 (pg/mL)	2.8 ± 0.8	2.4 ± 0.3	2.7 ± 0.9	2.3 ± 0.5	2.8 ± 0.6	3.1 ± 0.9
IL-10 (pg/mL)	11.5 ± 8.9	18.7 ± 11.1 *^§^	10.1 ± 3.8	15.0 ± 4.5 *^§^	13.8 ± 6.5	5.6 ± 4.0 *
TNF-α (pg/mL)	3.8 ± 1.7	2.9 ± 0.5 *^§^	4.0 ± 2.3	2.9 ± 0.6 *^§^	3.2 ± 1.7	3.3 ± 1.9
CRP (mL/dL)	3.1 ± 1.6	2.3 ± 1.4 *^§^	3.1 ± 1.2	2.2 ± 0.6 *^§^	3.0 ± 2.1	4.4 ± 2.1 *
TRAPP (µmolTrolox)	887 ± 187	883 ± 154	875 ± 134	844 ± 199	902 ± 163	898 ± 89
AOPP (µmol/L)	93.0 ± 30.4	77.5 ± 13.9 *^§^	88.1 ± 13.9	75.0 ± 13.5 *^§^	75.9 ± 17.8	86.3 ± 19.3
FOX (mmol/L)	0.45 ± 0.23	0.27 ± 0.07 *^§^	0.49 ± 0.29	0.30 ± 0.11 *^§^	0.48 ± 0.31	0.57 ± 0.34
NOx (µmol/L)	7.3 ± 2.6	7.4 ± 2.8	7.4 ± 2.1	7.0 ± 2.4	7.8 ± 4.1	8.3 ± 4.1

Note: MJ-SJ = group that performed RT in a multi-joint to single-joint order, SJ-MJ = group that performed RT in a single-joint to multi-joint order, CO*N* = control group, TC = total cholesterol, HDL-c = high-density lipoprotein cholesterol, LDL-c = low-density lipoprotein cholesterol, VLDL-c = very low-density lipoprotein cholesterol, IL-6 = interleukin-6, IL-10 = interleukin-10, TNF-α = tumor necrosis factor alpha, CRP = C-reactive protein, TRAPP = transport protein particle, AOPP = advanced oxidation protein products, FOX = ferrous oxidation-xylenol orange, NOx = nitric oxide metabolites. * *p* < 0.05 vs. pre-training; ^§^
*p* < 0.05 vs. control group at post-training.

**Table 2 ijerph-20-01165-t002:** Effect size values of the pre- to post-intervention changes for analyzed variables.

	MJ-SJ (*n* = 15)		SJ-MJ (*n* = 14)		CO*N* (*n* = 15)	
**Body composition**						
Body mass	0.03	(Trivial)	−0.06	(Trivial)	0.02	(Trivial)
Fat mass	−0.17	(Trivial)	−0.17	(Trivial)	0.06	(Trivial)
Relative fat mass	−0.43	(Small)	−0.35	(Small)	0.09	(Trivial)
Trunk fat	−0.14	(Trivial)	−0.21	(Small)	0.02	(Trivial)
Relative trunk fat	−0.42	(Small)	−0.44	(Small)	0.05	(Trivial)
**Blood biomarkers**						
Glucose	−1.34	(Large)	−1.66	(Large)	0.13	(Trivial)
Total cholesterol	0.04	(Trivial)	−0.36	(Small)	0.39	(Small)
HDL-c	0.08	(Trivial)	0.17	(Trivial)	−0.17	(Trivial)
LDL-c	−0.10	(Trivial)	−0.45	(Small)	0.52	(Moderate)
VLDL-c	−0.33	(Small)	0.00	(Trivial)	0.11	(Trivial)
Triglycerides	0.32	(Small)	0.03	(Trivial)	−0.18	(Trivial)
IL-6	−0.52	(Moderate)	−0.52	(Moderate)	0.39	(small)
IL-10	1.13	(Large)	0.77	(Moderate)	−1.28	(Large)
TNF-α	−0.47	(Small)	−0.58	(Moderate)	0.05	(Trivial)
CRP	−0.49	(Small)	−0.55	(Moderate)	0.86	(Large)
TRAPP	−0.02	(Trivial)	−0.19	(Trivial)	−0.02	(Trivial)
AOPP	−0.75	(Moderate)	−0.63	(Moderate)	0.50	(Moderate)
FOX	−0.65	(Moderate)	−0.69	(Moderate)	0.33	(Small)
NOx	0.03	(Trivial)	−0.14	(Trivial)	0.17	(Trivial)

Note: MJ-SJ = group that performed RT in a multi-joint to single-joint order, SJ-MJ = group that performed RT in a single-joint to multi-joint order, CO*N* = control group, HDL-c = high-density lipoprotein cholesterol, LDL-c = low-density lipoprotein cholesterol, VLDL-c = very low-density lipoprotein cholesterol, IL-6 = interleukin-6, IL-10 = interleukin-10, TNF-α = tumor necrosis factor alpha, CRP = C-reactive protein, TRAPP = transport protein particle, AOPP = advanced oxidation protein products, FOX = ferrous oxidation−xylenol orange, NOx = nitric oxide metabolites.

## Data Availability

The raw data supporting the conclusions of this article will be made available by the corresponding author, without undue reservation.

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
