# Peer review of "Effect of Resistance Exercise Order on Cardiovascular Disease Risk Factors in Older Women: A Randomized Controlled Trial"

_ijerph, 2023, doi:10.3390/ijerph20021165_

Round 1

Reviewer 1 Report

Line 40. The sentence does not make sense.  I believe the word "been" could be removed.

Did the authors use any strategies for participants to attend the exercise sessions?

Were there any differences in the groups once the data were compared after the dropouts?  What are the characteristics of the participants that dropped out?  Is an intention to treat analysis appropriate?

Did participants engage in 100% of sessions over the 12-week program?

Line 211 - the word without is misspelled (whithout)

Reviewer 2 Report

The design of this study is nota appropriate, not enough to be a RCT.

Reviewer 3 Report

Dear Authors,

I believe the paper has good scientific soundness and it is relevant.

However, the introduction section must be extended with more previous works to be cited.

Conversely, the Discussion section is a bit long and not easy to read. I would then suggest shortening it more synthetically.

Minor notes are present in the pdf file.

Best Regards

Round 2

Reviewer 3 Report

Dear Authors,

thank you for following my suggestions. I believe the paper has now been improved.